

# A new gigantic carnivore (Carnivora, Amphicyonidae) from the late middle Miocene of France

Floréal Solé[1], Jean-François Lesport[2], Antoine Heitz[3] and Bastien Mennecart[3]

[1] Royal Belgian Institute of Natural Sciences, Brussels, Belgium
[2] Private, Sainte-Hélène, France
[3] Naturhistorisches Museum Basel, Basel, Switzerland

## ABSTRACT

Serravallian terrestrial vertebrates are very uncommon in the northern margin of the Pyrenean Mountains. A mandible of a new large sized amphicyonid (ca. 200 kg) is here described from the marine deposits of Sallepisse (12.8–12.0 Mya). Despite that this new taxon is close in size to some European amphicyonids from the Miocene (e.g., *Amphicyon*, *Megamphicyon*, and *Magericyon*), the unique morphology of its p4, unknown in this clade, allows the erection of the new genus *Tartarocyon cazanavei* nov. gen. & sp. This taxon may be derived from a *Cynelos*-type amphicyonine. The description of this new taxon highlights the erosion of the ecological and morphological diversity of the Amphicyonidae in response to well-known Miocene events (i.e., Proboscidean Datum Event, Middle Miocene Climatic Transition, Vallesian Crisis).

## INTRODUCTION

The middle Miocene (15.97–11.63 Ma) is a period of great interest concerning climate change and faunal dispersal in Eurasia and Africa (*Rögl, 1999*; *Hilgen, Lourens & van Dam, 2012*). The Langhian (ca. 15.97–13.65 Mya) encompasses the Middle Miocene Climatic Optimum, a global increase in temperature of ca. 5 °C, while during the Serravallian, cooler temperatures occurred (*Hilgen, Lourens & van Dam, 2012*). These events led to important environmental changes and faunal renewals and exchanges (*Costeur, 2005*). Despite the very abundant invertebrate fossil record, little is currently known about the faunal connections between the northern and southern part of the Pyrenees Mountain range during the middle Miocene due to a lack of continental vertebrate remains. Indeed, the Southwestern part of France was flooded by the sea several times during the early and middle Miocene (*Cahuzac et al., 1992*) and the continuing uplift of the Pyrenees formed a natural barrier between the Iberian Peninsula and the rest of Europe.

The last transgression in the Aquitaine occurred during the Serravallian (middle Miocene, ca. 13.82–11.63 Mya). This sea deposited in the Orthez area (Southwestern France) a famous and abundant marine fauna found in shelly sandy deposits named "Faluns bleus" (*Delbos, 1848*), also known as Blue Faluns of Orthez (*Lesport, Cluzaud & Verhecken, 2015*). This formation attracted scientists early in paleontological history.

Corresponding author
Bastien Mennecart,
mennecartbastien@gmail.com

In 1833, the naturalist Dufour made an excursion in this area (*Dufour, 1836*) and gave indications to his palaeontologist friend Grateloup who soon after published new fossil gastropod species (*Grateloup, 1835*; *Grateloup, 1845-1847*). Since then, numerous authors have contributed to the knowledge of the malacofauna from the Orthez area, including in Sallespisse (see *Lesport, Cluzaud & Verhecken, 2012*; *Lesport, Cluzaud & Verhecken, 2015* for an extensive literature review). These bioclastic accumulations (thanatocenoses) may represent a nearshore environment in a subtropical to tropical climate. In 1993, JFL and Philippe Renard found a mandible of a very large carnivoran in a transgressive microconglomerate layer from the Crousquillière locality in Sallespisse. It was, at that time, the only terrestrial remain among the entire fauna in this layer. This specimen belongs to an amphicyonid (Carnivora, Caniformia).

The Amphicyonidae, which are colloquially referred to as "bear-dogs", represent one of the most characteristic groups of carnivorans in the Miocene European faunas (*Solé et al., 2018*). They first appeared during the Eocene (Priabonian, MP18, ca. 37–36 Ma; *de Bonis, 1978*; *Solé et al., 2018*). Nevertheless, the Miocene is particularly interesting for studying the evolution of this family. These carnivorous mammals included numerous species during the early and middle Miocene in Europe (*Viranta, 1996*), but went extinct before the end of the Miocene, the last European amphicyonids being known from the late Tortonian (*Amphicyon pannonicus*; *Kretzoi, 1985*; *Viranta, 1996*). Late Miocene amphicyonids are characterized by the presence of a pronounced, trenchant dentition (*Morlo et al., 2020*; *Morales et al., 2021a*), but during the early and middle Miocene there are other, less carnivorous forms (*Ginsburg, 1999*).

Three subfamilies of Amphicyonidae are recognized in the Miocene of Europe: the Haplocyoninae, the Thaumastocyoninae, and the Amphicyoninae, which are supposedly paraphyletic (*Morales et al., 2021b*). The typical haplocyonines (*Haplocyon*, *Haplocyonoides*, and *Haplocyonopsis*) are unknown in Europe after MN3 (*Peigné & Heizmann, 2003*; *Morlo et al., 2020*)—although they might have survived until the end of the Serravallian in Asia (*Jiangzuo et al., 2021*). Based on phylogenetic analysis, *Jiangzuo et al. (2021)* proposed to include in the Haplocyoninae the genera *Sarcocyon*, *Gobicyon*, and *Aktaucyon*. Among these genera, only *Gobicyon* is known from Europe (*G. serbiae* in MN6; *Pavlovic & Thenius, 1959*; *Ginsburg, 1999*; *Jiangzuo et al., 2018*). The Thaumastocyoninae groups the genera *Thaumastocyon*, *Ysengrinia*, *Tomocyon*, *Crassidia*, *Agnotherium*, *Ammitocyon*, and possibly *Amphicyonopsis* (*Morales et al., 2019*; *Morales et al., 2021a*; *Morales et al., 2021b*; *Morlo et al., 2020*). The Amphicyoninae as defined by *Peigné et al. (2008)* is now considered to probably be paraphyletic, forming a grade and including several lineages more basal than the thaumastocyonines or included in this subfamily (*Morales et al., 2019*; *Morales et al., 2021a*; *Morales et al., 2021b*). *Morales et al. (2021b)* created two new tribes (Pseudarctini and Magericyonini) to clarify the systematics of Miocene amphicyonines. Amphicyonini groups the genera *Amphicyon*, *Cynelos*, *Euroamphicyon*, *Heizmannocyon*, *Megamphicyon*, and *Paludocyon*—we here use the genus *Megamphicyon* but see *Morales et al. (2021b)* and *Van der Hoek et al. (2022)* for opposite opinion regarding the validity of this genus. Pseudarctini groups the genera *Ictiocyon*, *Dehmicyon*, and *Pseudarctos*. Magericyonini comprises the hypercarnivorous genus *Magericyon* and with some doubt *Pseudocyon*.

European Miocene amphicyonids were also ecologically diverse: taxa ranged in body mass from 9 kg to 320 kg and displayed typical mesocarnivorous, omnivorous, bone-crushing, and hypercarnivorous diets (*Viranta, 1996*; *Ginsburg, 1999*). They started to decline from MN7/8 with only a few taxa recorded during MN9-MN12 (*Viranta, 1996*). The amphicyonids may have suffered from the Vallesian Crisis, with only rare and specialized taxa known in the late Vallesian and early Turolian in some parts of Central Europe (*Agustí, Cabrera & Garcés, 2013*; *Viranta, 1996*). Therefore, the description of this new Amphicyonidae from Serravallian of Southwestern Europe grants novel insight into understanding the diversity and geographic distribution of the last amphicyonids and their abrupt decline in Europe.

## Geological settings and location

**Location and paleontological content.** During the Serravallian, the sea expanded into the gulf of Chalosse (Southwestern France), which was delimited by the "Diapir de Dax", the "Ride de Tercis", and the "Dôme de Clermont", and the anticline of Louer, and penetrated further south, constituting the Gulf of Orthez/Salies-de-Béarn (Fig. 1). The Blue Faluns in the area of Orthez are found in many places, mainly in the South part of Sallespisse, at an altitude comprised of 120 and 140 m (Le Paren, Houssé, Pouchan, Labarthe, Carré; see *Karnay, 1997*). All these localities are in line with a southwest/northeast orientation. The proximity and a global similarity in the taxonomic composition of the fauna and the sedimentological content allowed previous authors to consider all these localities as synchronous and they were grouped under the locality name of Sallespisse (*Daguin, 1948*). Nevertheless, very small differences in proportions within the different mollusc communities are observed, indicating small local environmental differences (*Degrange-Touzin, 1895*). The most common gastropod families are the Naticidae, Epitoniidae, Ocenebrinae, Nassariidae, Cancellariidae, Conidae, Turridae, and Acteonidae, which for the most part are predators, scavengers, or commensals. Among many species of bivalves, the most common genera are *Acanthocardia, Megacardita, Anadara, Pecten*, and *Clausinella*. These bivalves and the profusion of a species of scaphopod collected in a soft bioclastic sand matrix currently live on a sandy-muddy bottom of the SFBC type ("Sables Fins Bien Calibrés" = fine sands well calibrated, *Peres & Picard, 1964*). The current SFBC biocenosis, which occupies large areas along the coasts and bottom of the Mediterranean gulf, are remarkable for the absence of algae and marine phanerogams, which seems to agree with the deposits at the Carré site. This is confirmed by the abundant associated marine life (*e.g.*, *Nolf & Steurbaut, 1979*; *Chaix & Cahuzac, 2005*). However, some brackish and freshwater species (*e.g.*, *Theoxodus*) may indicate sediments of continental origin. The locality of Crousquillière (Fig. 1), misspelled in *Lesport, Cluzaud & Verhecken (2015)* as La Croustillère, is located on the Carré farm property (also known as Carrey) owned by the Cazanave family in Sallespisse. The fossiliferous Blue Faluns, grey-blue sands may be found along a small stream that flows into a brook called Le Moussu, south to the Carré farm (coordinates 43.512705; −0.717866). This locality was poorly exploited for its fossiliferous remains before the 1990s. From 1993, J-F Lesport and P Renard systematically excavated numerous fossils from these layers (crustaceans, bryozoans, echinoderms, foraminifers,
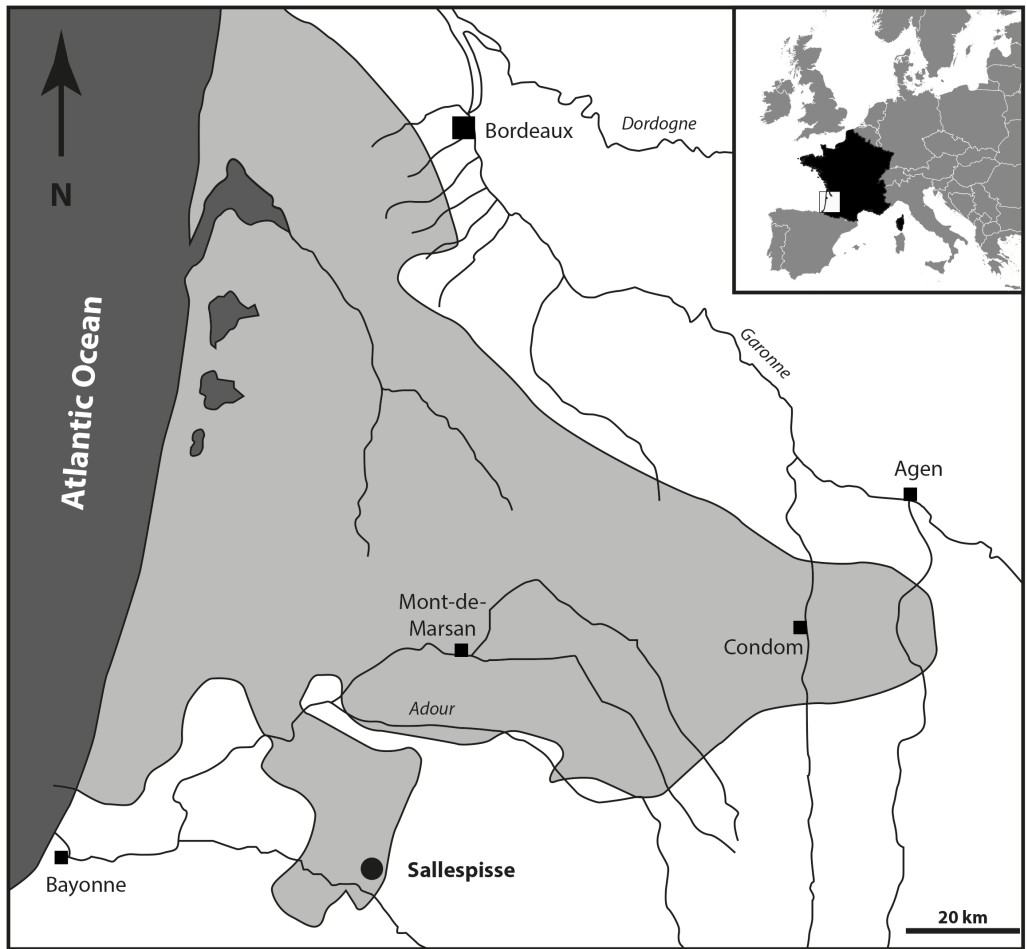

**Figure 1** **Geographical position of the fossiliferous locality of Sallespisse (Close-up on the Southwest France, redrawn from *Cahuzac, Janin & Steurbaut, 1995*).** The light grey area represents the maximum of extension of the Serravallian Sea.

scleratinians, fishes, and more than 200 species of molluscs; *Lesport, Cluzaud & Verhecken, 2015*). A new excavation campaign during the summer of 2021 completed the malacofauna but unfortunately did not bring new bone elements from carnivorous mammals.

**Sedimentological succession** (Fig. 2). The succession is relatively similar to the one observed in the other Blue Faluns outcrop from Sallespisse. The studied outcrop measures 3.5 m. It is composed from base to top of:

- Molasse deposits observed represent more than 10 m all along the stream. They are made of continental/lacustrine, whitish to greyish marly limestone with nodules. These sediments are apparently azoic. Nevertheless, the broad sedimentation of this molassic formation may be dated between the middle Eocene and the Burdigalian in this area (*Karnay, 1997*). Being at the very end of this sequence may indicate an age between the late Oligocene and the early Miocene. The top of this formation is heterogeneous, incised by shallow depressions forming a small bowl (ca. 1 meter in depth).

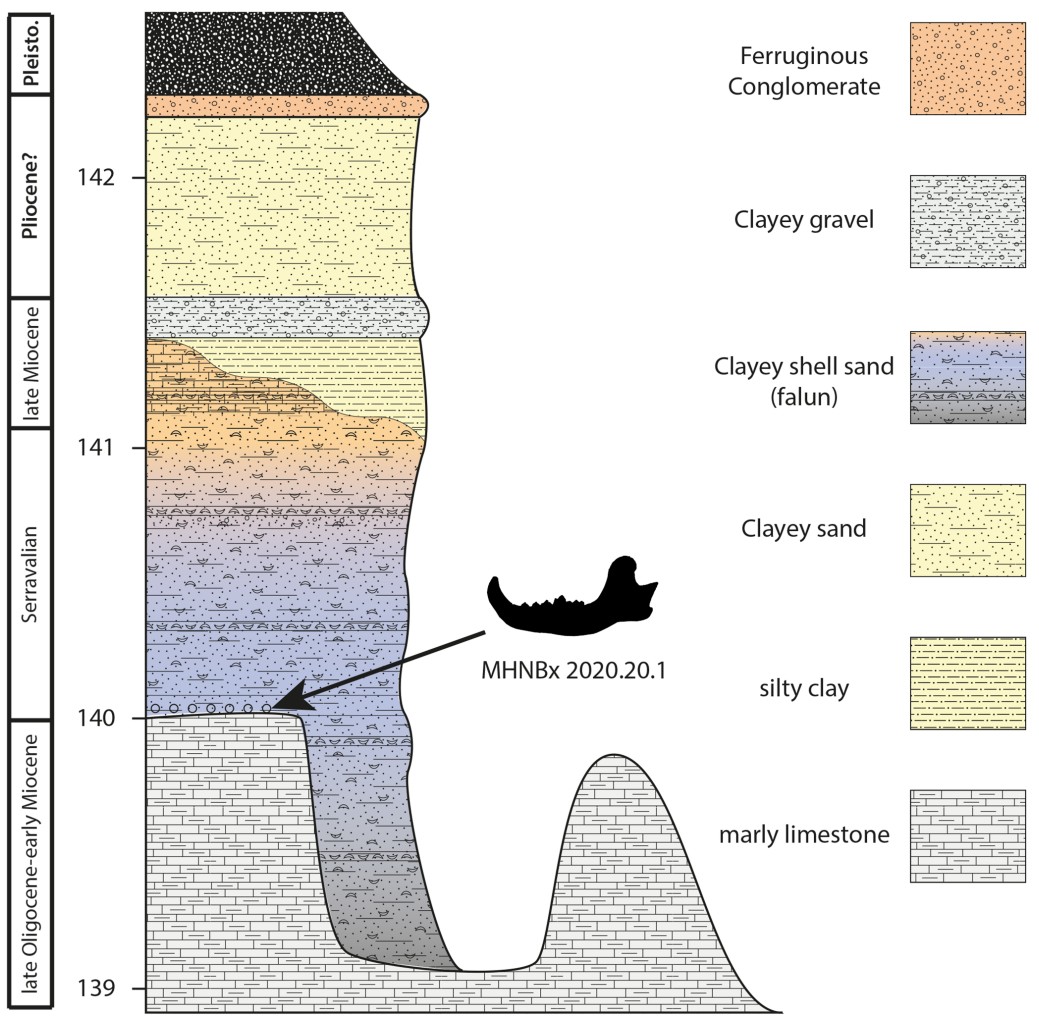

**Figure 2  Sedimentological succession of the Sallespisse outcrop with the location of the specimen MHNBx 2020.20.1.**

- Blue Faluns of Orthez (1 to 2 m) deposits with a variation of colour and sedimentation from base to top. The basal transition between the molasse deposits and the falun deposits is marked by broken molluscs and black pebbles that may be pierced by lithophagous bivalves, characteristic of a transgressive event. The studied mandible was found in this layer. New remains (an isolated molar and an astragalus) of a ruminant and cetaceans coming from this layer are currently under study. The basal basins are filled with blue to black clayey sand containing a diversified fauna of large molluscs (*e.g.*, *Pelecyora*, *Procardium*, *Megacardita*, *Trunculariopsis*, *Conus*). This level is sealed with a few centimetres of fine blue to black sand containing rare fossils. Then, the grey-blue falun has a thickness of ca 1 meter, containing many well-preserved mollusks. The Faluns deposits end with a yellow to orange sandstone characteristic of oxidating conditions. This Formation clearly corresponds to the Faluns de Sallespisse (*Karnay, 1997*). The age of these deposits is discussed below.

- A multicolored clay layer of 20 cm is found above the Faluns deposits. The top of the layer ends with fine ferruginous sandstone (2 cm), also called garluche. Lignified wood remains have been found during excavation in this section.

- Coarse yellowish clay sand (80 cm) ending with a ferruginous conglomerate (ca. 10 cm) that may correspond to Pliocene deposits. *Daguin (1948)*, without differentiating the different terrestrial levels, calls this formation "Sables Fauves".

**Age of the la Crousquillière (in Sallespisse) locality.** The age of the Falun deposits in the Orthez area have been interpreted many times variously as from the late Eocene (*d'Orbigny, 1852*) to the late Miocene (*Delbos, 1848*; *Raulin, 1852*), including an early Miocene age (*Grateloup, 1845-1847*). Nevertheless, the very diverse mollusc fauna permits constraining the age attribution of these deposits to the middle Miocene, characterizing the lithofacies Vindobonian (*Poignant, 1967*); the Sallomacian, a local name for middle Miocene marine deposits (*Fallot, 1893*; *Poignant, 1967*; *Nolf & Steurbaut, 1979*); or the sedimentological facies "Helvetian", which encompasses the Langhian and Serravallian (*Benoist, 1884*; *Degrange-Touzin, 1895*; *Cossmann & Peyrot, 1909–1924*; *Peyrot, 1925–1935*; *Peyrot, 1927–1932*). *Magné, Gourinard & Wallez (1987)*, *Cahuzac & Poignant (1993)*, and *Karnay (1997)* proposed a Langhian age for these deposits. However, recent studies based on diverse marine fauna (benthic foraminifers, ostracods, pteropods) and strontium isotopic analyses have led to a revaluation of the age of the Faluns deposits from Sallespisse and Orthez to the Serravallian (*Cahuzac, Janin & Steurbaut, 1995*; *Cahuzac & Poignant, 1996*; *Ducasse & Cahuzac, 1997*; *Cahuzac & Janssen, 2010*). These sediments are now attributed to the marine biozones Martini NN6/7, Blow N11/13, Janssen & King NSB19, with an isotopic age between 12.8 and 12.0 Mya. This corresponds to the European Land Mammal Ages MN7/8 (*Duranthon & Cahuzac, 1997*).

# MATERIALS & METHODS

**Specimen, nomenclature and measurements.** The specimen has been donated by JFL to the Natural History Museum of Bordeaux (France): it is now registered under the number MHNBx 2020.20.1. A cast of the specimen is available at the Natural History Museum Basel. Moreover, MHNBx 2020.20.1 has been surface scanned. The 3D model of the specimen is downloadable from the open access article *Mennecart et al. (2022)*.

The LSID for this publication is: urn:lsid:zoobank.org:pub:9FE7C271-9402-4062-B9B5-2087C8ACDC04. The online version of this work is archived and available from the following digital repositories: PeerJ, PubMed Central SCIE and CLOCKSS.

The dental nomenclature of premolars follows *Ginsburg (1999)*. The measurements, taken by calipers, have an accuracy of 0.1 mm.

**Body Mass.** We used the equation of *Van Valkenburgh (1990)* for all Carnivora irrespective of familial assignment in order to estimate the body mass of some amphicyonids including MHNBx 2020.20.1: $Log^{10}(BM) = [2,97 \times Log^{10}(Lm1)] -2,27$; with BM: the estimated body mass in kg; Lm1: the length of the first lower molar in millimeters.

**Biochronology.** The biostratigraphic framework is based on geological time scales for the Miocene provided by *Hilgen, Lourens & van Dam (2012)*.

## Systematic Palaeontology

Order CARNIVORA *Bowdich, 1821*
Suborder CANIFORMIA *Kretzoi, 1943*
Family Amphicyonidae *Trouessart, 1885*
Tribe Amphicyonini *Trouessart, 1885*
Genus *Tartarocyon* nov. gen.
***ZooBank LSID***. urn:lsid:zoobank.org:act:70359DC0-49E9-4E87-BC90-B02D5CFAFBB1

**Type species.** *Tartarocyon cazanavei* nov. gen. & sp.; monotypic, see below.

**Etymology.** Tartaro is the name of a legendary man-eater giant living in the Southwestern French Pyrenees, including the Bearn where the fossil has first been described. –*cyon* is the Greek for dog.

**Diagnosis.** As for the type and only species.

Species *Tartarocyon cazanavei* nov. gen. & sp.

Figure 3

***ZooBank LSID***. urn:lsid:zoobank.org:act:C7BE021C-6434-4715-AB89-63E9A64E6178

**Etymology.** Dedicated to Mr Alain Cazanave, owner of the locality, who helped with the excavation during many years.

**Diagnosis.** Large size Amphicyoninae possessing a complete dental formula. The taxon is characterized by the following features: long diastemata between the premolars, low p2 and p3, absent anterior accessory cuspid on p4, large and individualized distal accessory cuspid on p4, and unreduced m2 and m3. The taxon differs from all the European amphicyonids from the Miocene by the individualization of the distal accessory cuspid from the main cuspid on p4 and the extreme reduction of the distal shelf and cingulid.

**Specimen.** MHNBx 2020.20.1, right mandible bearing p2-p4, alveoli of i1-i3, c, p1, m1-m3.

**Measurements.** Tables 1 & 2.

**Description.** The mandible is mesiodistally elongated. Large diastemata are present between the canine, p1, p2, p3, and p4; the longest diastema is between the p2 and p3. The symphysis is oval and nearly horizontally oriented; it is high and extends posteriorly up to the distal root of p2. A mental foramen lies beneath the p1-p2 diastema; it is in a high position on the mandibular ramus. The ramus of the mandible is shallower anteriorly than posteriorly, the highest portion being below the m3. The ventral margin of the ramus below the toothrow is relatively straight, but beneath the anterior extremity of the large, deep masseteric fossa it becomes convex. An incisura vasorum is present on the ventral margin of the mandible anterior to the angular process. The angular process is robust but very short; it projects medially. The mandibular condyle is at the level of the tooth row. It is cylindrical and mediolaterally elongate. The coronoid process is tall and distinctly oriented backwards; it arises at a 50° angle relative to the horizontal ramus. The posterior margin of the coronoid is vertical and straight, while the cranial margin is rounded. The masseteric fossa, on its labial side, is deep and wide. The mandibular foramen is relatively circular, standing at the level of the incisura vasorum, at mid-height between the base of the

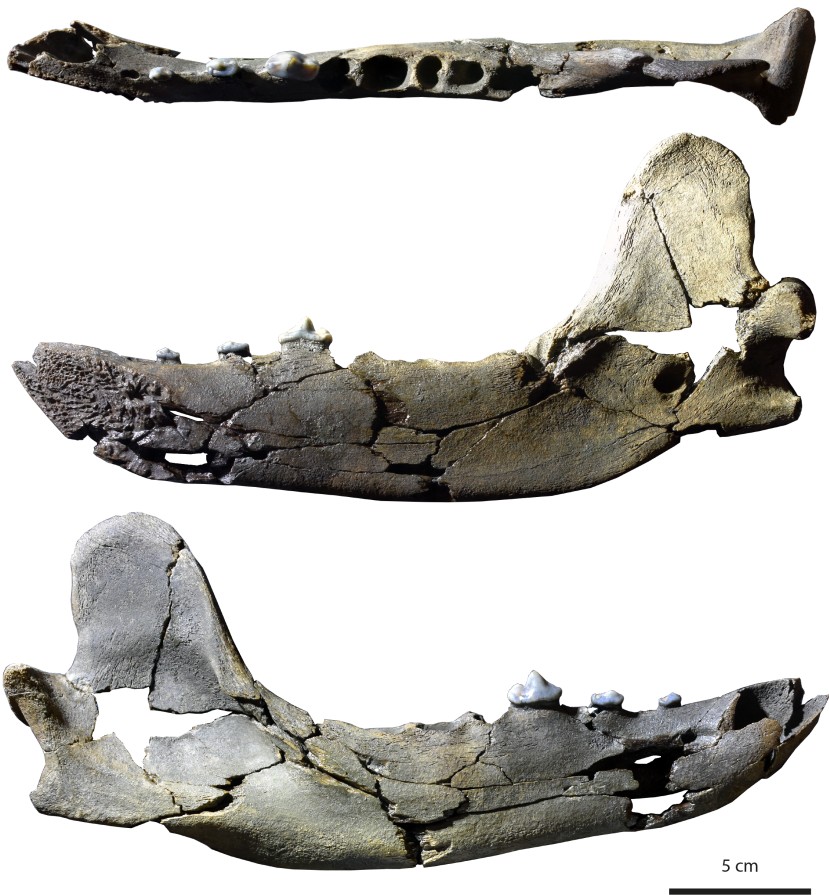

**Figure 3** Holotype (MHNBx 2020.20.1) of *Tartarocyon cazanavei* nov. gen. & sp. from Sallespisse (MN7/8, Southwest France), in occlusal, lingual, and labial views. Scale bar is 5 cm.

mandible and the level formed by the toothrow. The mandibular foramen opens midway between the m3 and the mandibular condyle.

The lower incisors are not preserved, but the alveoli of the i1, i2, and i3 are visible. Considering the size of the tooth sockets, the i3 seems to have been the largest and the i1 the smallest. The canine is also not preserved. It was ovoid in section and of large size. Its root extends in the mandible to between p2 and p3. The p1 is not preserved; a single alveolus is visible, but it appears that two, mainly fused, roots were present. The other teeth are two-rooted, except the m3, which is single-rooted. The p2 and p3 are very low in height. There is a prominent ridge on the mesial and distal margins of the main cuspid of these teeth. The main cuspid is low and located mesially, which results in an asymmetric morphology in lateral view. Mesial to the main cuspid, the lingual cingulid is thicker, but no individualized anterior cuspid is present. On p3 and p4, the distal shelf forms the widest part of the crown; it is less clear on p2. There is a short distal cingulid, but no cuspid is present. The p4 is distinctly longer and mediolaterally wider than the p2 and p3. However, the main cuspid remains low. The tooth is less asymmetric, the apex of the main cuspid

**Table 1** Measurements of the teeth of the holotype (MHNBx 2020.20.1) of *Tartarocyon cazanavei* nov. gen. & sp. from Sallespisse (MN7/8).

| Tooth locus | Length | Width |
| --- | --- | --- |
| i1 | 7.58[*] | 3.19[*] |
| i2 | 9.88[*] | 5.02[*] |
| i3 | 11.51[*] | 5.15[*] |
| c | – | 18.02[*] |
| p1 | 7.87[*] | 3.86[*] |
| p2 | 8.27 | 4.63 |
| p3 | 11.14 | 6.35 |
| p4 | 18.58 | 9.67 |
| m1 | 34.30[*] | 13.88[*] |
| m2 | 24.26[*] | 14.22[*] |
| m3 | 17.21[*] | 11.93[*] |

Notes.
[*]based on alveoli.

**Table 2** Several measurements of the teeth and mandible of the holotype (MHNBx 2020.20.1) of *Tartarocyon cazanavei* nov. gen. & sp. from Sallespisse (MN7/8).

| | |
| --- | --- |
| Length p1-p4 | 69.94 |
| Length m1-m3 | 78.67 |
| MD below p2 | 39.69 |
| MD below m1 | 48.97 |
| MD below m3 | 53.25 |

Notes.
MD, Mandible height.

being more mesiodistally centered. No real anterior accessory cuspid is present mesial to the main cuspid. A distal accessory cuspid is present: it is mostly individualized from the main cuspid. The distal accessory cuspid is mediolaterally centered. The distal cingulid is thin on the labial and lingual parts and is almost completely absent at the distal part; it does not form a distal shelf. The molars are not present, but the m1 was the largest tooth of the tooth-row. The m2 is larger than the m3.

**Comparison.** The premolars of the typical haplocyonines (*Haplocyon*, *Haplocyonoides*, *Haplocyonopsis*; *de Bonis, 1966*; *Peigné & Heizmann, 2003*; *Morlo et al., 2020*) differ from those of MHNBx 2020.20.1 in being high (*i.e.,* high main cuspid) and in the loss of the p4 distal accessory cuspid. Like the typical haplocyonines, the premolars of *Gobicyon serbiae* (MN6) differ from those of MHNBx 2020.20.1 in being high. Moreover, the p2 and p3 of *G. serbiae* possesses an individualized and large distal accessory cuspid. Additionally, typical haplocyonines and *Gobicyon* have a short toothrow lacking diastemata. These amphicyonids are thus relatively short-snouted compared to the taxon from Sallespisse.

All the thaumastocyonines differ from MHNBx 2020.20.1 in having relatively shorter diastemata between the premolars. The p2 and p3 preserved on MHNBx 2020.20.1 are similar to those of the oldest thaumastocyonines (*Ysengrinia*, *Crassidia*) in being low (*i.e.,*

their main cuspid is noticeably lower than the p4 main cuspid). The p4 of MHNBx 2020.20.1 also shares with the thaumastocyonines the presence of a strong distal accessory cuspid (Fig. 4); the youngest thaumastocyonines (*e.g., Agnotherium, Ammitocyon*) shares with the p4 of MHNBx 2020.20.1 the reduced distal shelf and cingulid (Fig. 4). However, the p4 of the thaumastocyonines differs from that of MHNBx 2020.20.1 in having a leaning backward p4 main cuspid (Fig. 4). The youngest thaumastocyonines—*Ammitocyon* and *Agnotherium*—moreover, differ from MHNBx 2020.20.1 in having no p1, p2, and p3 (*Morlo et al., 2020*; *Morales et al., 2021a*). Compared to the fossil from Sallespisse, the thaumastocyonines have a reduced m3 relative to m1; the youngest thaumastocyonines (*Thaumastocyon. Ammitocyon, Agnotherium*) have even reduced m2 relative to m1 in addition to lacking m3 (*Morlo et al., 2020*; *Morales et al., 2021a*). As a consequence, MHNBx 2020.20.1 differs in having more developed premolars, a mesially elongated snout (*i.e.,* diastemata between the premolars), and less reduced postcarnassial molars.

Three amphicyonines are regarded to be separate from those recorded in the Miocene: *Ictiocyon, Dehmicyon,* and *Pseudarctos* (*Ginsburg, 1992*; *Morales et al., 2021b*). They are all included among Pseudarctini (*Morales et al., 2021b*). These small amphicyonids are short-snouted (*i.e.,* short diastemata between the premolars) and the p2 and p3 are distinctly taller than on MHNBx 2020.20.1. Moreover, the distal accessory cuspid on p4 is reduced to lost in *Dehmicyon, Ictiocyon,* and *Pseudarctos* (*Ginsburg, 1992*; *Morales et al., 2021b*) (Fig. 4).

The hypercarnivorous *Magericyon* (*Peigné et al., 2008*), which belongs to the tribe Magericyonini (*Morales et al., 2021b*) differs from MHNBx 2020.20.1 in the absence of p2, in having a single-rooted p3, a p4 relatively shorter compared to the m1 (Table 3) and in the absence of a distal cuspid on p4 (Fig. 4). The genus *Pseudocyon* is probably close to *Magericyon* according to *Morales et al. (2021b)*. MHNBx 2020.20.1 is similar to the species of *Pseudocyon* in the presence of very long diastemata between the premolars and of low p2, p3. However, the p4 is relatively mesiodistally shorter (compared to the m1) in the *Pseudocyon* species than in MHNBx 2020.20.1; moreover, the distal part of the p4 of *Pseudocyon* is widened compared to that of the p4 of MHNBx 2020.20.1 (Fig. 4).

The Miocene Amphicyonini *Cynelos, Amphicyon, Megamphicyon, Euroamphicyon, Paludocyon,* and *Heizmannocyon* share with MHNBx 2020.20.1 the presence of very long diastemata between the premolars, the presence of low p2, p3, and p4, and the unreduced m3 (the m3 indeed tends to reduce and is even absent in hypercarnivorous amphicyonids; Table 3) (*Kuss, 1965*; *Peigné & Heizmann, 2003*; *Viranta, 1996*). Despite sharing a characteristically slender ramus of the mandible, the p4 of MHNBx 2020.20.1 differs from that of the *Cynelos* species by the absence of an anterior accessory cuspid (even if this structure is not individualized in *Cynelos*) and a much more reduced distal shelf (Fig. 4). The case of *Cynelos* is interesting because its p4 does not display a widening of its distal part; in this regard, its p4 is similar to that of MHNBx 2020.20.1 in occlusal view (Fig. 4). MHNBx 2020.20.1 shares with the species of *Paludocyon, Amphicyon, Heizmannocyon, Megamphicyon,* and *Pseudocyon* the reduction of the anterior accessory cuspid compared to *Cynelos*. However, the distal shelf of the p4 is more developed in these amphicyonines than in MHNBx 2020.20.1 and none of the above-mentioned species has a

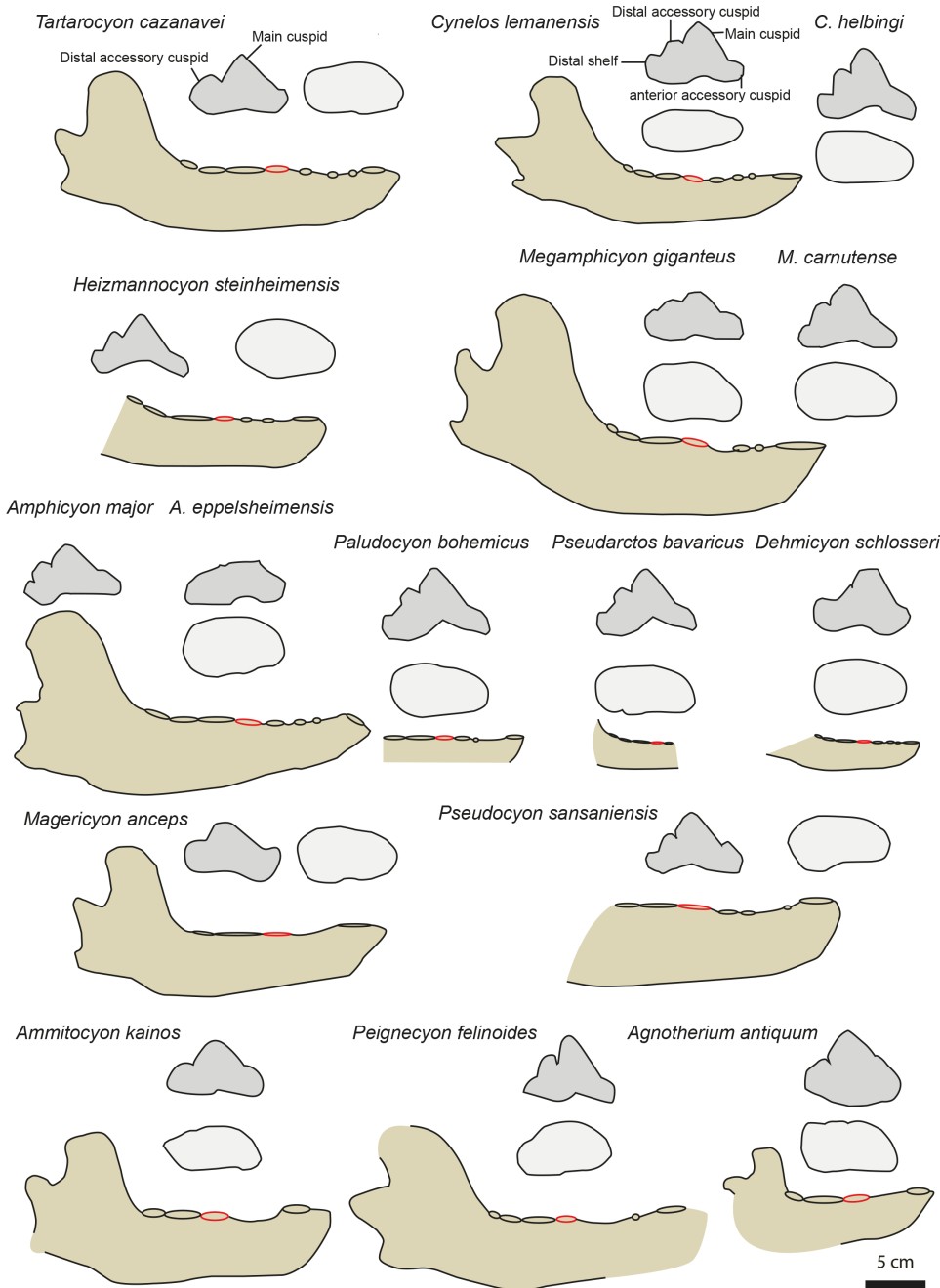

**Figure 4 Mandible and p4 comparison for several European amphycionids.** The red circle indicates the p4 position on the mandible. Modified from *Dehm (1950)*, *Kuss (1965)*, *Bergounioux & Crouzel (1973)*, *Viranta (1996)*, *Peigné & Heizmann (2003)*, *Peigné et al. (2008)*, *Nagel, Stefen & Morlo (2009)*, *Morales et al. (2021a)* and *Morales et al. (2021b)*. NMB TD1162 (*Heizmannocyon steinheimensis*), NMB SO4377 (*Megamphicyon giganteus*). The scale bar is five cm for the mandibles. The p4 are not to scale.

**Table 3** Ratios estimated based on premolars and molars for several amphicyonines and thaumastocyonines known in the Miocene of Europe.

| Taxon | Stratigraphic distribution | Ratio Lp2/Lm1 | Ratio Lp3/Lm1 | Ratio Lp4/Lm1 | Ratio Lm2/Lm1 | Ratiov Lm3/Lm1 |
|---|---|---|---|---|---|---|
| *Cynelos lemanensis* MNHNL-La85 | MN1-MN2 | 0.43 | – | 0.67 | 0.63 | – |
| *Crassidia intermedia* SMNS 46684 | MN1-MN2 | 0.47 | 0.43 | 0.63 | 0.58 | 0.38 |
| *Ysengrinia gerandiana* FSL 213828 | MN1-MN2 | 0.44 | 0.47 | 0.62 | – | – |
| *Cynelos rugosidens* BSP-1881-IX-14, 581 | MN2 | – | – | 0.67[*] | 0.65 | 0.42 |
| *Peignecyon felinoides* TU 7391147 | MN3 | – | – | 0.55 | 0.49 | – |
| *Megamphicyon carnutense* Fs 6953 | MN3 | 0.35 | 0.53 | 0.59 | 0.71 | – |
| *Cynelos helbingi* BSP-II-1937-12293 | MN3-MN4 | – | – | 0.57[*] | 0.64 | 0.39 |
| *Ictiocyon socialis* *Ginsburg* (*1992*, p. 311) | MN3-MN4 | 0.41 | 0.51 | 0.62 | 0.72 | 0.42 |
| *Ysengrinia depereti* MSNO.785 | MN3-MN4 | 0.25 | 0.43 | 0.48 | 0.62 | 0.34 |
| *Dehmicyon schlosseri* BSP 13562 | MN3-MN5 | 0.37 | 0.48 | 0.61 | 0.59 | 0.37 |
| *Paludocyon bohemicus* NM-PV 11723 | MN3-MN5 | 0.43 | 0.49 | 0.59 | 0.65 | 0.37 |
| *Pseudocyon sansaniensis* MNHN.F.Sa207 | MN3-MN9 | 0.28 | 0.29 | 0.51 | 0.6 | – |
| *Tomocyon grivense* UCBL-FSL 213797 | MN3-MN9 | – | – | – | 0.6 | – |
| *Megamphicyon giganteus* Specimen from Vienna & Basel SO6521 (*Hunt Jr, 2003*, table 4.7) | MN4-MN7/8 | 0.3 | 0.42 | 0.58 | 0.71 | – |

**Table 3** (*continued*)

| Taxon | Stratigraphic distribution | Ratio Lp2/Lm1 | Ratio Lp3/Lm1 | Ratio Lp4/Lm1 | Ratio Lm2/Lm1 | Ratiov Lm3/Lm1 |
|---|---|---|---|---|---|---|
| *Thaumastocyon bourgeoisi* Cast MNHN | MN5 | ? | ? | – | 0.45 | No m3 |
| *Pseudocyon steinheimensis* SMNS 4808 | MN5-MN7/8 | – | – | 0.44 | 0.64 | – |
| *Pseudarctos bavaricus* Ginsburg (1992, p. 309) | MN5-MN7/8 | – | – | 0.61 | 0.71 | 0.61 |
| *Amphicyon major* MNHN.F.Sa844 | MN6-MN9 | 0.31 | 0.36 | 0.54 | 0.7 | 0.56 |
| *Tartarocyon cazanavei* MHNBx 2020.20.1 | MN7/8 | 0.24 | 0.32 | 0.54 | 0.71 | 0.5 |
| *Amphicyon eppelsheimensis* Holotype | MN9 | – | – | 0.47 | 0.67 | – |
| *Magericyon castellanus* LVF 206y | MN9 | No p2 | – | 0.42 | 0.45 | – |
| *Agnotherium antiquum* NMB CM 242 & MNHM Epp 117-2017 | MN9/10 | No p2 | No p3 | 0.62 | 0.37 | No m3 |
| *Ammitocyon kainos* BAT-3′08 604 | MN10 | No p2 | No p3 | 0.71 | 0.54 | No m3 |
| *Magericyon anceps* Mean | MN10 | No p2 | 0.15 | 0.38 | 0.54 | – |

**Notes.**

Grey font, Thaumastocyonina; white font, Amphicyoninae.

p4 that displays a distal accessory cuspid separated from the main cuspid as it is on the p4 of MHNBx 2020.20.1. Moreover, these amphicyonine genera (see *Megamphicyon carnutense* and *Paludocyon bohemicus* in *Morales et al., 2021b*) possess a p4 that is characterized by a widening of the distal part. Additionally, the mandible of *Amphicyon* and *Megamphicyon* appears more massive than that of MHNBx 2020.20.1 (*Kuss, 1965*; *Peigné & Heizmann, 2003*; *Viranta, 1996*; Fig. 4).

A canine has been described from the locality of Rimbez (France, MN5), a locality that is located 100 km to the north-west of Sallespisse (*Ginsburg, 1967*); this locality is the closest one that has provided a Miocene amphicyonid specimen. This canine has been referred to *Pseudocyon sansaniensis*, an Amphicyonidae of similar size to MHNBx 2020.20.1. It is at the moment impossible to compare this canine with MHNBx 2020.20.1, but one can note that this tooth is close in size to the alveolus of the canine of MHNBx 2020.20.1. Despite

that the canine is much older than MHNBx 2020.20.1, one can imagine that the taxon from Rimbez could also be closely related to the taxon from Sallespisse.

To conclude, the fossil from Sallespisse shows striking similarities with the much older and clearly smaller *Cynelos*, a genus not yet known from the Middle Miocene of Europe (*i.e.,* presence of long diastemata between the premolars, unreduced premolars and m3, low p2 and p3, no widening of the distal part of the p4). The general morphology of the p4 remains relatively stable within the Amphicyoninae having a distal accessory cuspid more or less individualized and a distal shelf present (Fig. 4). MHNBx 2020.20.1 presents an unique morphology among the Amphicyoninae in having an individualized distal accessory cuspid on p4 and a distal shelf extremely reduced, extending the morphological range of the p4 in this subfamily (Fig. 4). Therefore, we erect the new genus and species *Tartarocyon cazanavei* nov. gen. & sp. for MHNBx 2020.20.1.

One can think that the basis for erecting a new taxon on this fragmentary material is not strong. However, because we are unable to discern which genus it most closely resembles and that given the originality of the morphology of the p4, we hypothesize that this specimen represents a new genus that will be further confirmed or not by future finds.

## DISCUSSION

**Relationships of *Tartarocyon cazanavei* nov. gen. & sp.** Because of the lack of information on the morphology of the molars, it is hard to discuss the relationships of *Tartarocyon cazanavei* nov. gen. & sp. within the amphicyonids; the molars actually provide numerous diagnostic features (see for instance the diagnoses in *Kuss, 1965*; *Viranta, 1996*; *Heizmann & Kordikova, 2000*; *Peigné & Heizmann, 2003*; *Peigné et al., 2008*; *Morales et al., 2019*; *Morales et al., 2021a*; *Morales et al., 2021b*). *Viranta (1996)*, *Peigné et al. (2008)*, *Morales et al. (2019)*, *Morales et al. (2021a)* and *Morales et al. (2021b)* tackled the relationships among European amphicyonids. However, the aims as well as the character and taxon lists used for the phylogenetic analyses are different in each analysis. Phylogenetic analysis of *Tartarocyon cazanavei* nov. gen. & sp. did not provide statistically significant results, adding noise to the topology forming polytomies, because the dentition of MHNBx 2020.20.1 is only represented by the p2, p3, and p4, including autapomorphic characters.

Nevertheless, as already highlighted, *Tartarocyon cazanavei* nov. gen. & sp. clearly differs from the Haplocyoninae, which possess tall and short premolars without diastemata. *Tartarocyon cazanavei* nov. gen. & sp. also does not belong to the Thaumastocyoninae, this family having reduced premolars and postcarnassial molars (Table 3). The youngest thaumastocyonine species, from the middle and late Miocene, are further characterized by the absence of m3 and of p1, p2, and p3, and a leaning backward main cuspid on p4 (Fig. 4, Table 3) (*Morales et al., 2019*; *Morales et al., 2021a*; *Morlo et al., 2020*). A reduction of premolar size is also observed in amphicyonines; this is a common trend in European amphicyonids. However, as seen on Table 3, the premolar and molar ratios show that the premolars (except the p4) and postcarnassial molars tend to reduce more among the thaumastocyonines than in the amphicyonines amphicyonini *Megamphicyon*, *Cynelos*, and *Amphicyon* (Table 3). The values estimated for *Tartarocyon* nov. gen. are similar to

those of *Cynelos*, *Megamphicyon*, and *Amphicyon* (Table 3). Moreover, diastemata are still present between the premolars in these amphicyonines as in *Tartarocyon cazanavei* nov. gen. & sp. Interestingly, the ratio between the p4 and the m1 is greater in the thaumastocyonines (except for *Ysengrinia depereti*, Table 3) than in *Megamphicyon*, *Amphicyon*, and *Tartarocyon* nov. gen.

The case of *Magericyon* is puzzling. This amphicyonid differs from the contemporaneous thaumastocyonines by the presence of an m3 but also by the presence of a reduced p4 compared to the m1 (Table 3) (*Peigné et al., 2008*; *Morales et al., 2019*; *Morlo et al., 2020*). In contrast, its shoulder anatomy is relatively primitive and generalized, being similar to that of *Cynelos lemanensis*. Its shoulder is intermediate between that of the ursid-like amphicyonines (*Amphicyon major*) and that of the markedly cursorial North American amphicyonids (Temnocyoninae and Daphoeninae) (*Siliceo et al., 2015*). *Morales et al. (2021b)* highlighted the originality of *Magericyon* in including this genus among the tribe Magericyonini. They also included, but with some doubt, the genus *Pseudocyon* in this tribe. One can note that this amphicyonine also has a reduced p4 compared to the m1 (Table 3).

It appears that *Tartarocyon cazanavei* nov. gen. & sp. is morphologically similar to *Cynelos*, *Amphicyon*, and *Megamphicyon* in having premolars and postcarnassial molars that are only slightly reduced in length. However, one can note that the anterior accessory cuspid area and the distal shelf are more reduced in *Tartarocyon cazanavei* nov. gen. & sp. compared to *Cynelos*. *Tartarocyon cazanavei* nov. gen. & sp. also differs from *Cynelos* by its reduced p2, p3, and p4 (Table 3), its late middle Miocene occurrence, and its much larger size. This feature is shared with *Amphicyon*, *Paludocyon*, and *Megamphicyon*. However, *Tartarocyon cazanavei* nov. gen. & sp. recalls *Cynelos* in having a p4 that does not show a widening of its distal part; at the opposite, *Amphicyon*, *Paludocyon*, and *Megamphicyon* have p4 that is characterized by a widening of the distal part. Despite these similarities, *Tartarocyon cazanavei* nov. gen. & sp. differs from *Cynelos* and *Amphicyon* in the large and individualised distal cuspid that is positioned distally on the p4; moreover, the distal shelf and distal cingulid is more reduced in *Tartarocyon cazanavei* nov. gen. & sp. than in *Cynelos* and *Amphicyon*. As a consequence, we think that *Tartarocyon cazanavei* nov. gen. & sp. is derived from a *Cynelos*-type amphicyonine.

*Cynelos* and *Amphicyon* are Amphicyonini known from the early Miocene (*Ginsburg, 1999*). *Tartarocyon* nov. gen. seems to be more derived than *Cynelos* but more basal than *Amphicyon*. *Tartarocyon cazanavei* nov. gen. & sp. followed a distinct evolutionary path from the other amphicyonids possibly due to geographic isolation, as shown by its unusual p4 morphology.

**Ecology of *Tartarocyon cazanavei* nov. gen. & sp.** The estimated body mass (based on the alveoli of the m1 of MHNBx 2020.20.1) is 194.91 kg. *Tartarocyon cazanavei* nov. gen. & sp. is distinctly larger than the species of *Cynelos*, which range from 13 to 86 kg (*Viranta, 1996*; Table 4). In being close to 200 kg, the estimated body mass of *Tartarocyon cazanavei* nov. gen. & sp. recalls those of *Amphicyon major* (212 kg, male), *A. pannonicus* (198 kg), *Magericyon* spp. (171–246 kg), and *Megamphicyon carnutense* (182 kg) (*Viranta, 1996*; Table 4). Amphicyonids that are significantly larger than *Tartarocyon cazanavei*

nov. gen. & sp. are few: *Megamphicyon giganteus* (317 kg, male), *A. gutmanni* (246 kg), *A. eppelsheimensis* (225 kg), *Magericyon castellanus* (246 kg), and *Amphicyonopsis serus* (270 kg) (*Viranta, 1996*; Table 4). In this regard, the amphicyonid from Sallespisse is one of the largest amphicyonids ever recorded in Europe.

*Viranta (1996)* recognized four categories of amphicyonids based on feeding ecology: omnivores, mesocarnivores, bone-crusher mesocarnivores, and hypercarnivores. The presence of the four premolars as well as the presence of large m2 and m3 (relative to the m1) indicate that *Tartarocyon cazanavei* nov. gen. & sp. was not a hypercarnivore. Indeed, hypercarnivorous amphicyonids such as *Magericyon castellanus*, *Pseudocyon caucasicus*, *Thaumastocyon* spp. and *Agnotherium* spp. are regarded as hypercarnivores because they are characterized by a reduction of the premolars and of the m2 and m3 together with the development of slicing carnassials (*i.e.,* P4 and m1) (*Viranta, 1996*). The high mass of *Tartarocyon cazanavei* nov. gen. & sp. contrasts with those of the omnivorous amphicyonids *Pseudarctos bavaricus* and *Ictiocyon socialis*, which were the smallest amphicyonids in the Miocene of Europe together with the mesocarnivorous *Dehmicyon schlosseri* (*Viranta, 1996*; *Morales et al., 2021b*). Moreover, the Pseudarctini *P. bavaricus*, *D. schlosseri*, and *I. socialis* are characterized by high-crowned teeth with blunt cuspids and closely appressed premolars; these two features distinguish these small amphicyonids from *Tartarocyon cazanavei* nov. gen. & sp. *Viranta (1996)* regarded *Cynelos* spp. as a typical mesocarnivore. This amphicyonid is notably characterized by a primitive dentition (*e.g.,* canine not especially robust, a crowded premolar row). *Tartarocyon cazanavei* nov. gen. & sp. clearly differs in having large diastemata between the premolars as well as a robust canine. *Viranta (1996)* considered *Amphicyon major* and *M. giganteus* as bone-crushing mesocarnivores. As noted by *Viranta (1996*, p.46), "There are no modern analogues for the dentitions of these species. They have well-developed molars and a sparsely distributed, complete set of premolars." These features are also found in *Tartarocyon cazanavei* nov. gen. & sp. Moreover, the body mass of *Tartarocyon cazanavei* nov. gen. & sp. and the *Amphicyon* species are close (see above). Therefore, *Tartarocyon cazanavei* nov. gen. & sp. can be reconstructed as a predator with bone-crushing habits (Fig. 5).

**The evolution of European amphicyonids during the Miocene.** *Viranta (1996)* carried out a comprehensive study on the systematics, ecology, and evolution of the European amphicyonids from the Miocene. The present discussion represents an update of the remarkable work of *Viranta (1996)* and underlines several periods to focus on.

*Viranta (1996)* did not consider the Haplocyoninae in her study. The inclusion of the Haplocyoninae, which were only present in the Miocene of Europe until MN3, reveals a similar specific diversity during the entire lower Miocene with 9 to 12 contemporaneous Amphicyonidae species in Europe (Table 5). The diversity seen in MN4 and MN5 is thus due to a diversification of the remaining amphicyonids (Amphicyoninae and Thaumastocyoninae), as already evidenced by *Viranta (1996)*, with a maximum of 11 species (in MN4) (Table 5).

Moreover, contrary to *Viranta (1996)*, the diversity of the Amphicyoninae and Thaumastocyoninae is already observed in MN3 (11 species in total; Fig. 6; Table 5). For instance, the locality of Tuchořice (Czech Republic) yielded one thaumastocyonine

**Table 4 List of the Amphicyonidae known in the Miocene of Europe with indication of their stratigraphic distribution, body mass, and diet.** Diet estimated based on similarities with the ones proposed by *Viranta (1996)*. The Haplocyoninae are here considered as hypercarnivores because they display a hypercarnivorous dentition (see *Wang, Wang & Jiangzuo, 2016*).

| Family-subfamily | Tribe | Taxon | Stratigraphic distribution | Body mass (in kg) | Diet |
|---|---|---|---|---|---|
| Amphicyoninae | Amphicyonini | *Amphicyon astrei* | MN1 | 112 | Bone-crushing mesocarnivores |
| | | *A. laugnacensis* | MN1-MN2 | 130 (est.) | Bone-crushing mesocarnivores |
| | | *A. lactorensis* | MN4-MN5 | 132 | Bone-crushing mesocarnivores |
| | | *A. major* | MN6-MN7/8 | 122–212[*] | Bone-crushing mesocarnivores[*] |
| | | *A. eppelsheimensis* | MN9 | 225 | Bone-crushing mesocarnivores |
| | | *A. gutmanni* | MN11 | 246[*] | Bone-crushing mesocarnivores[*] |
| | | *A. pannonicus* | MN11-MN12 | 198[*] | Bone-crushing mesocarnivores[*] |
| | | *Cynelos lemanensis* | MN1-MN2 | 42 | Mesocarnivores[*] |
| | | *C. rugosidens* | MN2 | 13 | Mesocarnivores[*] |
| | | *C. helbingi* | MN3-MN4 | 60-86[*] | Mesocarnivores[*] |
| | | *Euroamphicyon olisiponensis* | MN3-MN4 | 147[*] | Bone-crushing mesocarnivores[*] |
| | | *Heizmannocyon steinheimensis* | MN5-MN7/8 | 123[*] | Bone-crushing mesocarnivores[*] |
| | | *Janvierocyon pontignensis* | MN3 | 162 | Bone-crushing mesocarnivores |
| | | *Megamphicyon carnutense* | MN3 | 182 | Bone-crushing mesocarnivores |
| | | *M. giganteus* | MN4-MN7/8 | 157–317[*] | Bone-crushing mesocarnivores[*] |
| | | *Paludocyon bohemicus* | MN3-MN5 | 86 | Mesocarnivores |
| | | *Tartarocyon cazanavei* | MN7/8 | 195[**] | Bone-crushing mesocarnivores |
| | Magerocyonini | *Magericyon castellanus* | MN9 | 246 | Hypercarnivores[*] |
| | | *M. anceps* | MN10 | 171 | Hypercarnivores |
| | | *Pseudocyon sansaniensis* | MN3-MN9 | 126[*] | Bone-crushing mesocarnivores[*] |
| | | *P. caucasicus* | MN6 | 130[*] | Hypercarnivores[*] |
| | | *P. styriacus* | MN6 | 118[*] | Bone-crushing mesocarnivores* |
| | Pseudarctini | *Dehmicyon schlosseri* | MN3-MN5 | 23 | Mesocarnivores[*] |
| | | *Ictiocyon socialis* | MN3-MN4 | 21 | Omnivorous[*] |
| | | *Pseudarctos bavaricus* | MN5-MN7/8 | 9[*] | Omnivorous[*] |
| Thaumastocyoninae | | *Agnotherium antiquum* | MN9/10 | 148 | Hypercarnivores[*] |
| | | *Ammitocyon kainos* | MN10 | 120 | Hypercarnivores |
| | | *Crassidia intermedia* | MN1-MN2 | 169 | Hypercarnivores |
| | | *Amphicyonopsis serus* | MN6?-MN7/8 | 270 | Hypercarnivores |
| | | *Peignecyon felinoides* | MN3 | 110 | Hypercarnivores |
| | | *Thaumastocyon bourgeoisi* | MN5 | 72 | Hypercarnivores[*] |
| | | *T. dirus* | MN9 | 74 | Hypercarnivores[*] |
| | | *Tomocyon grivense* | MN3-MN9 | 174 | Hypercarnivores[*] |
| | | *Ysengrinia gerandiana* | MN1-MN2 | 72 | Hypercarnivores[*] |
| | | *Y. depereti* | MN3-MN4 | 118 | Hypercarnivores[*] |
| | | *Y. valentiana* | MN4 | 106 | Hypercarnivores[*] |

**Table 4** (*continued*)

| Family-subfamily | Tribe | Taxon | Stratigraphic distribution | Body mass (in kg) | Diet |
|---|---|---|---|---|---|
| Haplocyoninae | | *Gobicyon serbiae* | MN6 | 109 kg | Hypercarnivores |
| | | *Haplocyon crucians* | MN1-MN2 | 45 kg | Hypercarnivores |
| | | *H. elegans* | MN1-MN2 | 29 kg | Hypercarnivores |
| | | *Haplocyonoides mordax* | MN1-MN3 | 52 kg | Hypercarnivores |
| | | *H. suevicus* | MN2 | 42 kg | Hypercarnivores |
| | | *Haplocyonopsis crassidens* | MN1 | 85 kg | Hypercarnivores |

**Notes.**
*Bodymass and diet based on *Viranta (1996)*.

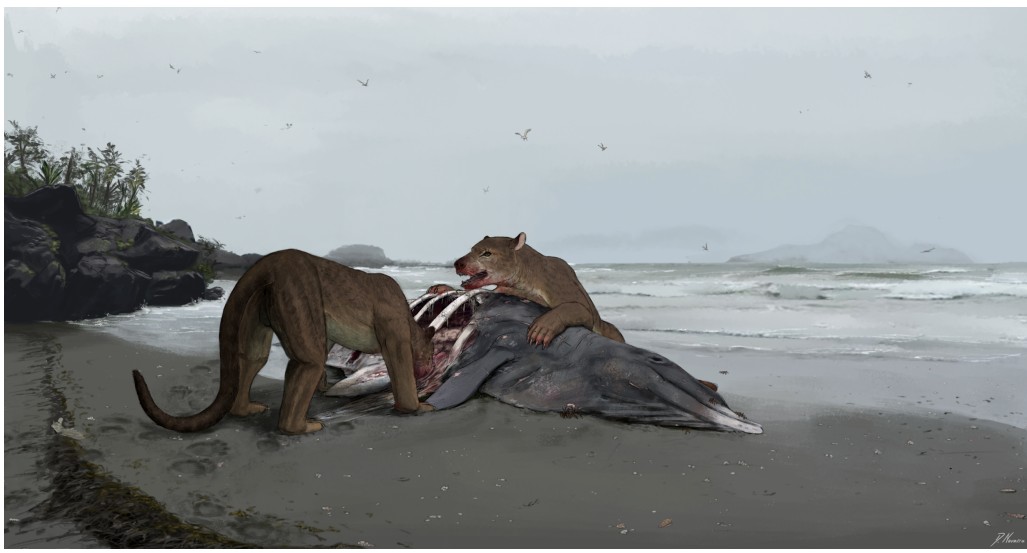

**Figure 5** **Reconstruction of *Tartarocyon cazanavei* nov. gen. & sp. feeding on a stranded dolphin along the Serravallian sea.** We know only few on the inland environmental conditions where *Tartarocyon* lived. This illustration thus combines all the data from the site la Crousquillière in Sallespisse including the intertidal dark deposits, the abundance of the molluscs, and the mandible of *Tartarocyon* in the high-tide line. Drawing by Denny Navarra.

(*Morales et al., 2019*) and three amphicyonines (two Amphicyonini and one Pseudarctini; *Morales et al., 2021b*). At the European level, the amphicyonids were clearly taxonomically and ecologically diverse in MN3 (Fig. 6; Table 5), as illustrated by the presence of the small omnivore *Ictiocyon*, the mesocarnivores *Cynelos* and *Dehmicyon*, the hypercanivore *Peignecyon*, and the large bone-crusher mesocarnivores *Pseudocyon*, *Amphicyon*, *Megamphicyon*, and *Janvierocyon*.

The diversification of the Amphicyoninae and Thaumastocyoninae must be questioned because it was concomitant with the disappearance of the Haplocyoninae (the last European haplocyonines are from MN3; *Peigné & Heizmann, 2003*). The MN3 biozone hosts some of the most important climatic and faunal events including the Proboscidean Datum Event and Asiatic dispersals (*e.g.*, *Tassy, 1989*; *Van der Made, 1999*). From arid environments throughout Western Europe during the Agenian, a latitudinal gradient developed, with

Table 5 Number of taxa by MN levels in totality and based on diet after Table 4.

| MN level | Omnivores | Mesocarnivores | Bone-crushing mesocarnivores | Hypercarnivores | Totality |
|---|---|---|---|---|---|
| MN1 | | 1 | 2 | 6 | 9 |
| MN2 | | 2 | 1 | 6 | 9 |
| MN3 | 1 | 3 | 4 | 4 | 12 |
| MN4 | 1 | 3 | 4 | 3 | 11 |
| MN5 | 1 | 2 | 4 | 2 | 9 |
| MN6 | 1 | | 5 | 4 | 10 |
| MN7/8 | 1 | | 5 | 2 | 8 |
| MN9 | | | 2 | 4 | 6 |
| MN10 | | | 0 | 3 | 3 |
| MN11 | | | 2 | | 2 |
| MN12 | | | 1 | | 1 |

wet and closed environments in France and Germany during the Orleanian (*Costeur, 2005*; *Costeur & Legendre, 2008*). Due to environmental restructuring and the competition from the newcomers, nearly 60% of the ungulate fauna was replaced during that time (*Scherler et al., 2013*). The restructuring of the community and of the environment may have been fatal to the Haplocyoninae but favored the Amphicyoninae and Thaumastocyoninae.

The amphicyonids remained diverse during MN5 (nine species), MN6 (10 species), MN7/8 (eight species), and MN9 (six species) (Table 5). The bone-crushing mesocarnivorous amphicyonids are taxonomically well-diversified in MN6 (five species) and MN7/8 (five species including *Tartarocyon* nov. gen.). On the other hand, mesocarnivorous amphicyonids are unknown in Europe after MN5. Additionally, amphicyonid between 50 kg and 100 kg are very rare after MN5: only *Thaumastocyon dirus* is known among this range (74 kg in MN9; Table 4) (Fig. 6). The disappearance of the mesocarnivorous amphicyonids and rarefaction of amphicyonids of 50–100 kg is related to the disappearance of *Cynelos* from Europe (Fig. 6; Table 5). One can, however, note the reappearance of the haplocyonines in MN6 (occurrence of *Gobicyon serbiae*; *Ginsburg, 1999*; *Jiangzuo et al., 2018*; *Jiangzuo et al., 2021*). This taxon probably dispersed from Asia into Europe because this genus appeared earlier in Asia (ca. 17 Ma; *Jiangzuo et al., 2021*) than in Europe. Interestingly, its mass (109 kg) is close to that of the amphicyonids known in MN6 and not to those of the *Cynelos* species recorded in MN5. Therefore, it did not probably fill the same ecological niche. Nevertheless, *Gobicyon* was present in Europe only for a short period and is only known from one locality (*Pavlovic & Thenius, 1959*; *Ginsburg, 1999*). A small reorganization of the amphicyonid fauna thus occurred between MN5 and MN6. This biotic event might be related to the Middle Miocene Climatic Transition (*Steinthorsdottir et al., 2021*), which results for instance in an increase in aridity in Spain (*Menéndez et al., 2017*).

From MN6 to MN11, the largest amphicyonids were all specialized as either hypercarnivorous or bone-crushing mesocarnivorous predators –except the case of the monospecific omnivorous *Pseudarctos*. The taxonomic diversity of the bone-crushing

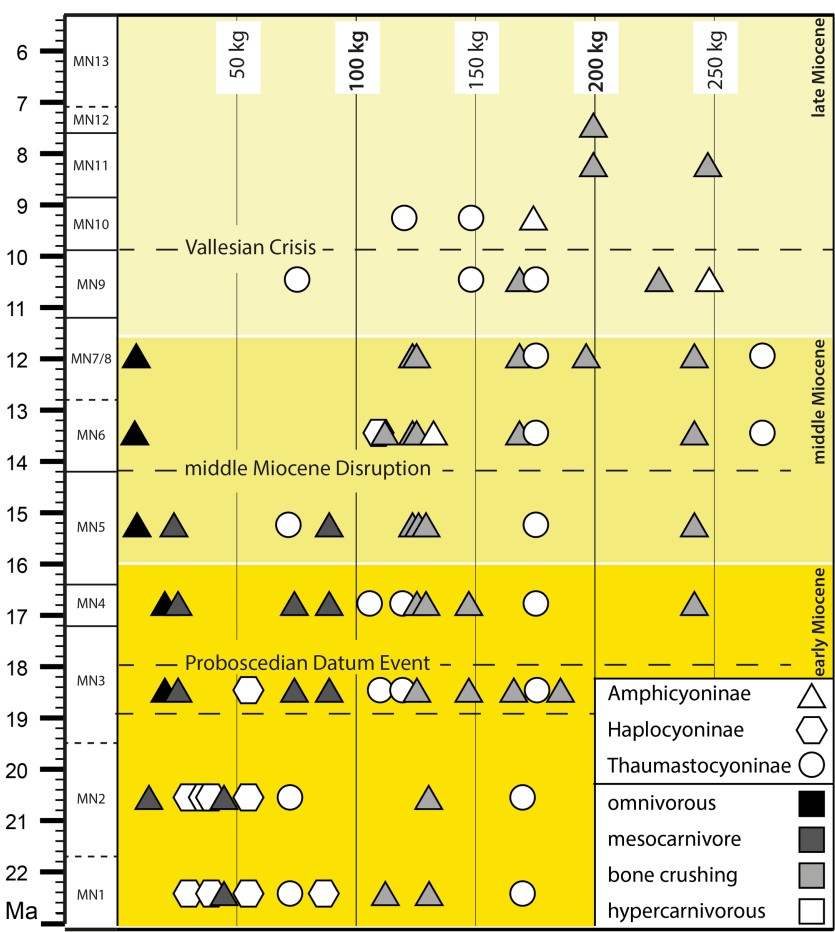

**Figure 6  Body mass and diet distribution of the amphicyonids during the Miocene biozones.** The horizontal dashed lines refer to the biotic events discussed in the text. The biostratigraphic framework follows *Hilgen, Lourens & van Dam (2012)*.

mesocarnivores is high in MN6 and MN7/8, but only two taxa are known in MN9 (Fig. 6; Table 5). In contrast, hypercarnivorous amphicyonids were still taxonomically diverse in MN9 with 4 species.

*Viranta (1996)* estimated that the decline of the Amphicyonidae started in MN7/8 and considered that MN9 marked the probable disappearance of amphicyonids in Western Europe. However, the recent descriptions of the amphicyonids *Magericyon anceps* (Magericyonini; *Peigné et al., 2008*), *Ammitocyon kainos* (Thaumastocyoninae; *Morales et al., 2021a*) in MN9 and MN10 Spanish localities, the first discovery of *Megamphicyon giganteus* in MN7/8 (Turkey; *Van der Hoek et al., 2022* and *Tartarocyon* nov. gen. have greatly changed our perception of the latest amphicyonid evolution (Fig. 6; Table 5). Indeed, the amphicyonids, notably the Thaumastocyonines, were still diverse in MN7/8 (eight species) and MN9 (six species) although less than in MN6. As noted by *Morlo et al. (2020)*, the Thaumastocyoninae reached its highest diversity in MN 9.

The omnivorous amphicyonid *Pseudarctos*, which was also the smallest and only omnivorous amphicyonid at that time (and last representative of the Pseudarctini), disappeared from Europe just before MN9 (last record in MN7/8; see *Morlo, Nagel & Bastl (2020)* for indication regarding a possible re-work of the fossils of *P. bavaricus* found at Eppelsheim) (Fig. 6; Table 5). As a consequence, the European amphicyonids are only represented by large to very large forms of at least 70 kg body mass during MN9, 100 kg during MN10, and even 200 kg during MN11 (Fig. 6). Moreover, this modification of the amphicyonid fauna also resulted in the presence of only specialized amphicyonids: the latter were either hypercarnivores or bone-crushing mesocarnivores (Table 5). To summarize, from MN9 to MN11, the amphicyonid community did not change considerably and abruptly (Fig. 6), but a trend, which is characterized by a disappearance of the smallest taxa together with a decrease of the diversity, is visible.

The slight modification of the amphicyonid fauna between MN9 and MN10 could be related to the Vallesian Crisis. This crisis coincided with the early/late Vallesian boundary (at 9.7 Ma) (Fig. 6). At first recognized in Spain (*Agustí & Moyà-Solà, 1990*; *Agustí, Cabrera & Garcés, 2013*), the Vallesian Crisis is now described as the major extinction event in the history of the Western European mammalian faunas (*Jaeger & Hartenberger, 1989*) (but see *Casanovas-Vilar et al., 2014* for a critical analysis). The Vallesian crisis was a time of major environmental change that led to a substantial turnover of mammals in Western Europe (*Fortelius et al., 1996*; *Agustí, Cabrera & Garcés, 2013*). The environmental change, notably characterized by an expansion of open habitats and retraction of forests, led to a decrease in the diversity of browsers. The opening of the environments led to the disappearance of small sized predators. This can thus explain the disappearance of *Thaumastocyon* from Europe.

Because *Viranta (1996)* extensively discussed the possible explanations for the decline of the amphicyonids (*e.g.*, extinction of potential prey, competition), we will not develop these discussions herein. *Agustí, Cabrera & Garcés (2013)* noted that the amphicyonids were affected by the Vallesian crisis and that only some poorly known amphicyonids persisted in the late Vallesian and early Turolian in some parts of Central Europe (*Amphicyon gutmanni* from Germany and Austria, and *Amphicyon pannonicus* from Hungary). Moreover, these amphicyonids were very large forms that display bone-crushing mesocarnivorous dentition (*Viranta, 1996*; Fig. 6). However, as mentioned above, the recent description of the hypercarnivorous amphicyonids *Ammitocyon* in a Spanish locality close to MN10 (*Morales et al., 2021a*) and *Magericyon* from Spanish localities close to MN9 and MN10 (*Peigné et al., 2008*) indicate that amphicyonids were still present in Southwestern Europe at the end of the Vallesian. Therefore, despite a decrease in number of species, amphicyonids remained present across Europe and display ecological diversity during MN10. However, as noted by *Viranta (1996)*, only the largest amphicyonids were still present in Europe at the end of the Vallesian and beginning of the Turolian. As a consequence, it appears that the Vallesian (not only the Vallesian crisis) seems to correspond to a period of decrease in diversity and in size range of bone-crushing mesocarnivorous and hypercarnivorous amphicyonids (Table 5). Because the decrease in taxonomic diversity is notable, the Vallesian period was not insignificant for the remaining amphicyonids.

## CONCLUSIONS

*Tartarocyon cazanavei* nov. gen. & sp. is a new large amphicyonid from the French locality Sallespisse (12.8–12.0 Ma, France). The specimen may represent a new genus that will be further confirmed or not by future finds. It clearly differs morphologically from the Thaumastocyoninae and Haplocyoninae. It seems that this amphicyonid is a part of the radiation of a group of amphicyonines during the Miocene after MN3 (as exemplified by the genera *Pseudocyon*, *Cynelos*, *Amphicyon*, and *Magericyon*); it probably derived from a *Cynelos*-type amphicyonine.

*Tartarocyon* nov. gen. illustrates the diversity of the amphicyonids in Europe: during MN7/8 amphicyonids were diversified in both the body mass and diet. However, the ecological and diversity reduction of the Amphicyonidae is polyphased. A new comprehensive analysis of the taxonomic and ecologic diversity of the amphicyonids is necessary to better understand the impact of biotic and abiotic factors on the evolution of these predators.

## ACKNOWLEDGEMENTS

Our thanks go to the Cazanave family and particularly to Alain, owner of the Carré farm for his welcome, authorisation and various information. Our gratitude also goes to Philippe Renard, friend and fellow excavator, who contributed to reviving our memories of Sallespisse, photos and additional information on the fauna collected. JM Pacaud (Muséum National d'Histoire Naturelle, Paris) is also thanked for his proofreading and suggestions. We also thank C. Gagnaison (Institut Polytechnique LaSalle Beauvais) for his comments regarding the amphicyonids from the Miocene and especially the mention of the canine from Rimbez. Laurent Charles and Nathalie Mémoire, curators in the Museum of Natural History of Bordeaux are also thanked. All our gratitude goes to the paleoartist Denny Navarra (d.navarra.work@gmail.com) for his drawings and patience. We are looking forward for future collaborations. BM would like to thank PeerJ for granting this article thanks to the PeerJ prize 2021 of the Association Paléontologique Française congress. We acknowledge the reviewers Lars Werdelin (Swedish Museum of Natural History) and Michael Morlo (Senckenberg Research Institute), the editor Brandon P. Hedrick (Louisiana State University Health Sciences Center).

### Funding
The authors received no funding for this work.

### Competing Interests
The authors declare there are no competing interests.
## Author Contributions

- Floréal Solé conceived and designed the experiments, performed the experiments, analyzed the data, prepared figures and/or tables, authored or reviewed drafts of the article, and approved the final draft.
- Jean-François Lesport performed the experiments, authored or reviewed drafts of the article, and approved the final draft.
- Antoine Heitz performed the experiments, authored or reviewed drafts of the article, and approved the final draft.
- Bastien Mennecart conceived and designed the experiments, performed the experiments, analyzed the data, prepared figures and/or tables, authored or reviewed drafts of the article, and approved the final draft.

## Field Study Permissions

The following information was supplied relating to field study approvals (i.e., approving body and any reference numbers):

The excavation was verbally approved by the land owner M Alain Cazanave, owner of the Carré farm.

## Data Availability

The 3D surface file of the specimen is available at MorphoMuseuM: 10.18563/journal. m3.163.

## New Species Registration

The following information was supplied regarding the registration of a newly described species:

Publication LSID: urn:lsid:zoobank.org:pub:9FE7C271-9402-4062-B9B5-2087C8ACDC04
Genus name: urn:lsid:zoobank.org:act:70359DC0-49E9-4E87-BC90-B02D5CFAFBB1
Species name: urn:lsid:zoobank.org:act:C7BE021C-6434-4715-AB89-63E9A64E6178.

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
