# Peer review of "A new gigantic carnivore (Carnivora, Amphicyonidae) from the late middle Miocene of France"

_PeerJ, doi:10.7717/peerj.13457_

## Round 0.1 · original submission · Major Revisions

Dear authors,

Thank you for your submission to PeerJ. Based on comments from two reviewers and my own reading, I believe that this manuscript will be publishable in PeerJ following major revisions.

Both reviewers note that the erection of a new genus is somewhat problematic and ask for additional discussion to justify the taxonomic decision. They also note that the paper would benefit from a thorough reading from a proficient English speaker. Both reviewers and I (see below) have made grammatical suggestions to increase readability. However, once these are made, having a native speaker check the resulting draft would be greatly beneficial.

When you submit your revision, please include a tracked changes version of your manuscript, a clean version, and a response to reviewer suggestions. Thank you again for your submission. Please contact me if you have any questions.

Best,

Brandon P. Hedrick, Ph.D.



Line 26: Reword ‘This group that is restricted to Europe seems…’

Line 42: ‘little is known about’

Line 45: ‘flooded several times’

Line 46: ‘and the Pyrenean Mountains that were continuing to rise, formed’

Line 52: ‘Early in paleontological history,’

Line 54: delete ‘could’

Line 55: ‘gastropod species’

Line 73: ‘Taxa ranged in body mass from 9 kg…’

Line 76: ‘suffered’

Line 93: Reword. ‘Similitude’ is an odd word here

Line 110: delete ‘may be found’ and comma after the )

Line 114: ‘owned’

Line 117: ‘1990s’. No apostrophe.

Line 125: delete ‘complete’

Line 135: ‘have an evolution of color’ would be grammatically correct. I don’t like the word evolution though, so perhaps reword.

Line 179: ‘registered’

Line 255: What do you mean ‘important in size’

Line 256: I’m not sure what you mean here. Is ‘expends’ the wrong word?

Line 287: ‘cusps’

Lin2 294: Shorter relatively or absolutely? Couldn’t the new taxon just be larger? This needs expansion

Line 337: ‘imagine’ rather than image

Line 356: What do you mean by statistically significant here?

Line 360: diastemata

Line 404: What do you mean by ‘under-estimated’ here?

Line 411: delete ‘well’. Maybe ‘much’?

Line 420: ‘amphicyonids’

Line 449: grammar

Line 468: ‘restructuring’

Line 471: ‘diverse’ rather than ‘diversified’

Line 514–515: ‘At first recognized…’

Line 536: ‘remained present across Europe…’

Line 539: ‘weighed’

Figure 3: ‘lingual’

Table 4: Two asterisks on Tartarocyon?

·

Basic reporting

During my reading I have made corrections to the English which, though perfectly comprehensible, could be improved for increased clarity and readability. My corrections are marked on the pdf.

All technical aspects as, as far as I can see, carried out in a professional manner.

Experimental design

This paper is descriptive and involves no experimental design. The paper ´clearly lays out what is new and why previous work on the topic needs to be updated.

Validity of the findings

This paper is composed of two distinct parts. The first is a description of a putative new Amphicyonidae taxon from the middle Miocene of southwest France. The second is a review of middle to early late Miocene amphicyonids from western Europe.

To begin with the second. This review is an update and broadening of that published by Viranta (1996). In the intervening years a number of new Amphicyonidae from the Miocene of Europe have been published. What makes the new review valuable is that the new taxa change a number of things related to amphicyonid ecology and extinction in the from the late middle Miocene onwards. I find this part of the paper well done and well worth publishing. However, Morales et al. (2021) recently published a new paper on amphicyonids of the late bearly Miocene of Czechia which should includes a rearrangement of taxa within Amphicyonini that should be added in revision to this manuscript regardless of agreement or disagreement over the generic assignments in that paper. Overall, however, this section of the manuscript requires only minor revision.

The other (first) part of the manuscript is the description of the new genus and species Tartarocyon cazanavei from the southwest France. This specimen (a mandibular ramus with a few teeth) is in my opinion interesting mainly because of its geographic provenance. I have no doubt that is a ‘local’ taxon, and it is certainly found on marine sediments. Nevertheless, I would like to see some evidence that the find site is not the result of substantial transport of a carcass down a river and into the sea, where it came to rest. If that were the case the environment where the animal lived may have been well inland. The in-life illustration, though very skillfully done (I do like it), seems to me rather fanciful in two ways. First, I see it as unlikely that an amphicyonid of this size would be hunting along the shoreline in that fashion. Lions and brown hyenas do along the Namib coast, but they are after slow-moving marine mammals for which there is no evidence here and no presence in the illustration. Second, the by now classic paper by Carbone et al. (1999) based on fundamental energetic requirements, precludes carnivores as large as this amphicyonid having hunted prey as small as the moschids shown in the illustration. I find it far more likely, regardless of the geological provenance of the specimen, that the actual animal would have been hunting large prey and eating carrion well inland.

Regardless of this, which may be seen as quibbling, I find the taxonomic conclusion in the paper problematic. This only a single, incomplete specimen and the taxonomic distinction made is based essentially on a single character – the configuration of the distal part of the p4. In a group of taxa in which dental variability is high (cf. the illustrations of Paludocyon bohemicus in Morales et al., 2021 for an example) this is very poor evidence for generic distinction (which should in any case be based on phylogeny not phenetics). That this is a new species is certainly possible and I would not argue with making it a new species of some known genus, perhaps as (e.g.) aff. Amphicyon cazanavei. It just seems to me that the generic splitting introduced by Ginsburg (in Rössner & Heissig, 1999) is (was) overdone but is now being pursued by Morales and colleagues. My recommendation is to either act on my suggestion above or provide a much better argument for generic distinction of this specimen, specifically adding more direct comparison with related taxa. I do not expect this to be problematic.

Refs.
CARBONE, C., MACE, G. M., ROBERTS, S. C. & MACDONALD, D. W. 1999. — Energetic constraints on the diet of terrestrial carnivores. Nature 402: 286-288.
MORALES, J., FEJFAR, O., HEIZMANN, E., WAGNER, J., VALENCIANO, A. & ABELLA, J. 2021. — The Amphicyoninae (Amphicyonidae, Carnivora, Mammalia) of the early Miocene from Tuchořice, the Czech Republic. Fossil Imprint 77: 126-144.
RöSSNER, G. & HEISSIG, K. 1999. — The Land Mammals of Europe. Verlag Dr. F. Pfeil.
VIRANTA, S. 1996. — European Miocene Amphicyonidae — taxonomy, systematics and ecology. Acta Zoologica Fennica 204: 1-61.

·

Basic reporting

The manuscript is well structured, well figured, and does present the new fossil in its temporal and spatial context. I nevertheless suggest to shorten the chapter on amphicyonid evolution in the European Miocene to the very few points, the new taxon adds information to.
The language could be improved. I gave some hints in the commented pdf, but I suggest review by a native English speaker.

Experimental design

Given the restricted knowledge on Miocene amphicyonids of Europe and the very uncommon geological circumstances the fossil was found in, a publication is more than welcome and well within the scope of PeerJ.
The uncommon morphology of p4 adds importance to this publication. All methods are state of the art, and the references are adequate and complete.

Validity of the findings

The authors base their new taxon on a single character, the morphology of the distal accessory cuspid of p4. In my view, it would be necessary to discuss the variability of p4 in the closest relatives (Amphicyon) to strengthen this taxonomic decision. On the other hand, all points containing discussions on thaumastocyonines and haplocyonines could strongly be shortened, as relative molar sizes exclude the new mandible to belong to these subfamilies.

Additional comments

If the manuscript concentrates on the comparison to Amphicyon and is shortened by information the new taxon does not reveal new evidence, it will be highly welcome and will attract a lot of readers.
Several minor suggestions can be found in the attached pdf.

---

## Round 0.2 · Minor Revisions

Dear authors,

Thank you for your attention to comments from a previous round of reviews. I am still not fully convinced of the justification for a new taxon, but the addition of Figure 4 is certainly quite helpful. Perhaps add in the argument that while the basis for erecting a new taxon on this fragmentary material is not strong, you are unable to discern which genus it most closely resembles and that given the stratigraphic and geographical importance of the specimen, you hypothesize that this specimen represents a new genus that will be further confirmed or not by future finds.

One reviewer re-examined this manuscript and had a number of additional comments that are in an attached pdf. In addition to those changes, I have added some line-by-line changes below that are primarily grammatical.

Please submit a tracked changes and clean version of your manuscript as well as a reviewer response document. Please let me know if you have any questions.

Best,

Brandon P. Hedrick, Ph.D.


Line by line:

Line 19: ‘vertebrates’

Line 26: Maybe ‘highlights the erosion of the ecological and morphological diversity of the…’

Line 27: What are the ‘well-known Miocene events’? Be specific here

Line 52: This citation format isn’t right for PeerJ. Please fix the 1845–1847. Also on line 176.

Line 59: amphicyonid

Line 85: Delete ‘whatsoever’

Line 98: Rather than ‘is crucial’, ‘grants novel insight into understanding the diversity…’

Line 108: remove the period before ‘(Figure 1)’

Line 148: uncapitalize ‘formation’ here

Line 151: ‘forming a small bowl’

Line 163: ‘characteristic of oxidating’ (or oxidizing?)

Line 174: ‘in the Orthez area’

Line 216: Do you mean ‘familial’?

Line 293–294: What do you mean by tall and short? Can you reword this and expand upon it? Also the next sentence repeats itself.

Line 365: ‘remains’. Delete ‘until now’

Line 367: ‘presents’

Line 369: ‘morphological range’

Line 385: ‘polytomies’

Line 464–466: ‘a crowded premolar row’

Line 503–504: ‘Due to environmental restructuring and…’

Line 539: ‘were still diverse in’

Line 577: ‘There are no taxa with masses below 150kg known after MN9”

Figure 4: ‘Mandible’ on line 849. Also ‘scale bar’ on Line 853.

Line 860: ‘mandible’

·

Basic reporting

All points are fine, exeptions are given as a comment in the pdf.

- A very new publication: DOI: 10.1007/s12542-022-00610-0 should be included
- Supplements with the references for the stratigraphical distributions would be helpful

Experimental design

no further comments

Validity of the findings

As in the first submitted version, the question to be debated is whether an uncommon p4 morphology is good enough to erect a new amphicyonid genus. In my view, one could probably separate a species.
Given the known variability within pseudarctine genera, I would avoid the erection of a new genus and follow the other reviewer in suggesting usage of aff. Amphicyon or Amphicyonini indet.
However, the added picture showing p4 morphologies among amphicyonids helps the reader to understand the position of the authors.
Everything else in this contribution is fine. Minor remarks can be found in the commented pdf.

---

## Round 0.3 · Minor Revisions

Dear authors,

Thank you for your corrections based on the previous round of review. I appreciate your adding the addition concerning the necessity of future finds to decisively confirm the validity of the species. I would like to you add a short sentence to that effect to the conclusions as well.

In addition, there were a few very minor additional grammar points that need to be fixed. After that, the paper will be publishable in PeerJ.

Thank you for your submission. Please contact me if you have any additional questions.

Best,

Brandon P. Hedrick, Ph.D.


Line 55: ‘for an extensive literature review)’

Line 79: remove comma after (2021)

Line 139: ‘contains’ isn’t the right word here. Do you mean ‘remains’?

Line 303: ‘leaning backward’ is odd phrasing. Perhaps ‘caudally oriented’ so long as that is what you mean.

Line 308: ‘relative to m1 in addition to lacking’

Line 397: Same odd phrasing as on line 303

Line 605: delete ‘moreover’

Fig 2 caption: change ‘location where the specimen’ to ‘location of the specimen’

Fig 3 caption: ‘lingual’

Fig 3 caption: ‘bar’ rather than ‘bare’

Tables 4 and 5 have some red text and strikethroughs. Make sure to get rid of that for publication.

---

## Round 0.4 · accepted · Accept

Dear authors,

Thank you for your submission to PeerJ. I would like to now move the paper on to the next stage and accept it for publication. Congratulations! Please let me know if you have any questions and I would be happy to answer them.

Best,

Brandon P. Hedrick, Ph.D.